# Stem Cell Models for Cancer Therapy

**DOI:** 10.3390/ijms23137055

**Published:** 2022-06-24

**Authors:** Nitin Telang

**Affiliations:** Cancer Prevention Research Program, Palindrome Liaisons Consultants, Montvale, NJ 07645-1559, USA; ntelang3@gmail.com

**Keywords:** breast and colon cancer, stem cells, testable therapeutic alternatives

## Abstract

Metastatic progression of female breast and colon cancer represents a major cause of mortality in women. Spontaneous/acquired resistance to conventional and targeted chemo-endocrine therapy is associated with the emergence of drug-resistant tumor-initiating cancer stem cell populations. The cancer-initiating premalignant stem cells exhibit activation of select cancer cell signaling pathways and undergo epithelial–mesenchymal transition, leading to the evolution of a metastatic phenotype. The development of reliable cancer stem cell models provides valuable experimental approaches to identify novel testable therapeutic alternatives for therapy-resistant cancer. Drug-resistant stem cell models for molecular subtypes of clinical breast cancer and for genetically predisposed colon cancer are developed by selecting epithelial cells that survive in the presence of cytostatic concentrations of relevant therapeutic agents. These putative stem cells are characterized by the expression status of select cellular and molecular stem cell markers. The stem cell models are utilized as experimental approaches to examine the stem-cell-targeted growth inhibitory efficacy of naturally occurring dietary phytochemicals. The present review provides a systematic discussion on (i) conceptual and experimental aspects relevant to the chemo-endocrine therapy of breast and colon cancer, (ii) molecular/cellular aspects of cancer stem cells and (iii) potential stem-cell-targeting lead compounds as testable alternatives against the progression of therapy-resistant breast and colon cancer.

## 1. Introduction

Progression of female breast and colon cancer to advanced-stage metastatic disease represents a major cause of mortality in the female population. The American Cancer Society projects combined new cases at 333,230 and cancer-related deaths at 68,060 in 2022 [1]. Conventional treatment options for breast cancer include cytotoxic chemotherapy, selective estrogen receptor modulators, selective estrogen receptor degraders, cyclin-dependent kinase inhibitors and aromatase inhibitors. Treatment options for colon cancer include cytotoxic multi-drug chemotherapy with DNA synthesis inhibitors, platinum-containing agents that generate cytotoxic DNA adducts, topoisomerase inhibitors, selective cyclooxygenase inhibitors and selective ornithine decarboxylase inhibitors [2].

Long-term treatment with these pharmacological agents is commonly associated with systemic toxicity, spontaneous/acquired tumor resistance and the emergence of therapy-resistant cancer stem cell populations. These major limitations associated with conventional chemo-endocrine therapy and small-molecule-based targeted therapy emphasize an unmet need to identify testable therapeutic alternatives against therapy-resistant breast and colon cancers. Natural products such as dietary phytochemicals, micro-nutrients and nutritional herbs have documented cancer-preventive efficacy on epithelial organ site cancers of the breast and colon, in part via targeting cancer stem cells [3,4]. These relatively nontoxic, naturally occurring agents, because of their documented human consumption, may represent potential alternatives.

The role of estrogens in breast cancer is well established as a promoter of hormone-responsive cancer subtypes. Estrogen receptor (ER) isoforms α and β belong to a superfamily of ligand-regulated nuclear transcription factors. Genomic signaling of ER involves a complex cascade of signaling events via the functioning of several co-activator/co-repressor molecules that culminates in the estrogen-response-element-mediated target gene expression of c fos and c jun. These signaling pathways are also involved in endocrine therapy resistance. ER-α signaling represents a predominant molecular mechanism for the positive growth regulation of breast cancer [5]. Additionally, the CYP-450-mediated generation of estradiol metabolites with distinct positive and negative growth-regulatory functions modulates the growth of breast cancer [6,7]. In contrast, ER-β exhibits transcriptional activity that is distinct from that of ER-α, and functions as a negative growth regulator [8,9,10,11]. However, the role of the ovarian steroid hormones in colon cancer is less defined. Published evidence suggests that estrogen receptors function as inhibitory modifiers for colon cancer. In the Apc ^MIN^/+ model, a lack of ER-α and ER-β accelerates the formation of colon cancer [12]. ER-β functions as a negative growth regulator in colon cancer and phytoestrogens function as potent ER-β agonists [13,14].

The normal stem cell population in epithelial organ sites is responsible for tissue regeneration and the preservation of homeostatic growth control via regulation of proliferation, differentiation and apoptosis. The Wnt/β-catenin, Hedgehog and Notch signaling pathways play critical regulatory roles in the maintenance, survival and function of the normal stem cell population [15,16]. In contrast, the cancer stem cells are characterized by the dysregulation of normal regulatory pathways, activation of survival pathways via phosphotidyl-inositol 3 kinase (PI3K), protein kinase B (AKT), molecular target of rapamycin (m TOR) and nuclear factor kB (NFkB) signaling and epithelial–mesenchymal transition (EMT) [17]. The well-established role of the cancer stem cell population in therapy-resistant disease progression emphasizes a focus on cancer stem cell models and the stem-cell-targeting efficacy of therapeutic alternatives. The cancer stem cells represent a minor subpopulation of the tumor that is responsible for therapy resistance and tumor initiation [18,19].

Several optimized assays for the isolation of putative cancer stem cells are available. These assays involve fluorescence-assisted cell sorting of Rodamine-123 positive side populations, cell-membrane-specific antibody positivity, drug-efflux-based assays, aldehyde dehydrogenase-1 (ALDH-1)-positive cells and the isolation of drug-resistant phenotypes. For the parental cell lines that provide a source of putative drug-resistant stem cells, mechanistic assays of the growth-inhibitory efficacy of natural products are widely used [20,21,22,23,24,25,26]. For the isolation of drug-resistant stem cells, the positive selection of resistant phenotypes in the presence of cytotoxic drug concentrations is effectively employed. The drug-resistant stem cells are characterized by the expression of stem-cell-selective tumor spheroid formation (cellular), cell surface molecules and nuclear transcription factors (molecular). The status of stem cell markers has been quantified by the extent of tumor spheroid formation and by the expression status of select molecules, such as clusters of differentiation CD44, CD133 and nuclear transcription factors octamer-binding transcription factor-4 (OCT-4), Kruppel-like factor-4 (Klf-4), sex determining region box Y-2 (Sox-2), cellular Myc (c-Myc) and DNA-binding transcription factor NANOG in cancer stem cell models [22,27,28,29]. Collectively, these stem cell markers provide specific and sensitive quantitative endpoints to characterize stem cell populations and also to confirm the stem-cell-targeted efficacy of test agents.

It is well recognized that stem cell populations in epithelial organ sites and in respective cancers play distinct and unique roles. The stem cell population is involved in tissue regeneration, organ site homeostasis and in the progression of the premalignant phenotype to a metastatic phenotype. The common and unique cellular and molecular characteristics of normal and cancer stem cells represent important concepts in stem cell biology. More importantly, cancer stem cell biology also facilitates the identification of mechanistic leads for the clinical translatability of the primary and secondary prevention of cancer and efficacious therapeutic intervention.

The overall objective of the present review is to discuss evidence-based concepts that are relevant to (i) cellular models for breast and colon cancer, (ii) the growth patterns of parental cell lines that contain putative stem cells, (iii) the growth-inhibitory efficacy of natural products (phytochemicals and nutritional herbs) on the developed models, (iv) the isolation and characterization of stem cell models developed from female breast and colon cancer-initiating target cells and (v) mechanistic leads for the efficacy of natural products on the developed cancer stem cell models. Collectively, this review provides a systematic analysis of conceptual and technical aspects and summarizes published evidence to provide a proof of concept that natural products may provide testable therapeutic alternatives for therapy-resistant breast and colon cancer.

## 2. Experimental Models

In the multi-factorial and multi-step carcinogenic process of sporadic breast and colon cancers, dietary, environmental and hormonal factors exert profound impacts on initiation, promotion and progression. Genetically predisposed patients carrying germ line mutations in the tumor suppressor *BRCA1* and *BRCA2* genes in breast and in the tumor suppressor *APC* gene in colon cancer may be at greater risk due to the effects of exogenous factors in developing cancer. Reliable in vitro models of relevant target organs for cancer provide valuable mechanistic approaches to identify efficacious cancer preventive agents and, thereby, complement in vivo investigations on animal models. From a therapeutic perspective, in vitro cellular models for drug-resistant stem cells facilitate investigations to identify testable alternatives against therapy-resistant breast and colon cancers.

Epithelial cell culture models represent facile in vitro experimental approaches to examine the multi-step process of the initiation, promotion and progression of organ site cancer. In addition, models developed from the non-tumorigenic target tissues provide valuable approaches to examine the direct cancer-initiating effects of prototypic organ-selective chemical carcinogens such as metabolism-dependent pro-carcinogens 7-12, dimethyl benz anthracene (DMBA), benzo [α] pyrene (BP), dimethyl hydrazine (DMH) and azoxy-methane (AOM), and metabolism-independent direct-acting methyl nitroso urea (MNU) and N-methyl-N-nitroso-guanidine (MNNG) [30,31,32,33], as well as of RAS, Myc and HER-2 oncogenes [34,35,36,37], on the carcinogenic process. Investigations focused on in vitro chemical- or oncogene-mediated carcinogenesis also provide facile experimental approaches to identify cellular and molecular endpoint biomarkers that are relevant to experimental modulations, specifically for the initiation, promotion and progression of the multi-step organ site carcinogenic process.

The parental cell lines developed from human breast carcinoma and from genetically predisposed mice exhibit hyper-proliferation, as evidenced by decreased population doubling time, increased saturation density, accelerated cell cycle progression and downregulated cellular apoptosis.

The data presented in Table 1 provide a summary of the molecular characteristics of various cellular models for breast and colon cancer.

The non-tumorigenic epithelial cells initiated for carcinogenesis by either chemical carcinogens, viruses or oncogenes exhibit AI growth in vitro and tumor formation in vivo. The tumorigenic status of the epithelial cells is confirmed by the in vivo tumor formation of transplanted cells. Thus, in the cell culture models, AI growth represents an in vitro surrogate endpoint marker for in vivo tumorigenic transformation that is indicative of a quantifiable risk of developing cancer [32,34,35,36,37].

The data presented in Table 2 illustrate the status of AI colony formation and tumor formation in the cellular models for breast cancer. The data in Table 3 illustrate the status of these endpoints in the cellular models for colon cancer. Comparison of these endpoint biomarkers in non-tumorigenic and tumorigenic epithelial cells demonstrates the specificity of these markers towards persistent tumorigenic transformation.

### 2.1. Mechanistic Assays

Several mechanistic assays that have been optimized for the parental cell lines and also for stem cells are widely used in preclinical cancer research. The assays for parental cell lines involve monitoring cell cycle progression, cellular apoptosis and the expression status of relevant regulatory molecules at genomic and proteomic levels. In the mechanistic assays for stem cells, tumor spheroid (TS) formation represents a commonly used biological marker. In addition, flow-cytometry-based assays have facilitated the quantitation of the expression status of additional stem cell markers. The quantitative immunofluorescence assay is optimized to measure the relative cellular uptake of antibodies specific for cell surface markers and for nuclear transcription factors. The cell surface markers include clusters of differentiation CD44 and CD133, while the molecular markers include nuclear transcription factors octamer-binding transcription factor-4 (OCT-4), Kruppel-like factor-4, (Klf-4), sex-determining region box Y-2 (SOX-2), cellular myc (c-myc) and homeobox transcription factor (NANOG) [22,27,28,29,38,39]. For the immunofluorescence assay, the cells stained with the fluorescein-labelled relevant antibodies are sorted using a flow cytometer and monitored for cellular fluorescence. The quantitative endpoint is represented by relative fluorescent units (RFU).

### 2.2. Pharmacological Agents

Conventional chemo-endocrine therapy for breast cancer includes the use of multi-drug cytotoxic chemotherapeutics containing doxorubicin, 5-fluro-uracil, cisplatin, paclitaxel, selective estrogen receptor modulators (SERM) such as tamoxifen, selective estrogen receptor degraders (SERD) such as fulvestrant, aromatase inhibitors such as letrozole and exemestane and cyclin-dependent kinase inhibitors such as palbociclib and ribociclib. For colon cancer, multi-drug combinations of 5-fluro-uracil+ irinotecan and 5-fluoro-uracil + oxaliplatin and non-steroidal anti-inflammatory drugs such as sulindac and celecoxib are used [22,27].

Some of these clinically relevant agents for breast and colon cancer are utilized to select the resistant phenotypes, as summarized in Table 4. Long-term treatment of cells with these agents at their respective maximum cytostatic concentrations eliminates the drug-sensitive phenotypes and facilitates the selective growth of drug-resistant phenotypes. Expansion of the surviving drug-resistant population in the presence of the drug provides the resistant phenotypes, representing putative stem cells for breast and colon.

## 3. Characterization of Cancer Stem Cells

Therapy-resistant cancer-initiating stem cells are characterized by the expression of select nuclear transcription factors, some of which are also documented to be crucial for the maintenance of drug-resistant stem cells and induced pluripotent stem cells. These stem-cell-specific transcription factors include OCT-4, Klf-4, SOX-2 and c-myc [38,39]. The expression statuses of the cellular and molecular stem cell markers are compared in the drug-sensitive and drug-resistant stem cells. The status of TS, CD44, NANOG and OCT-4 is enhanced in favor of the resistant population, and thereby provides a quantitative endpoint to characterize the drug-resistant TAM-R, LAP-R and DOX-R phenotypes.

### 3.1. Breast

The drug-resistant stem cells maintained in the presence of TAM, LAP and DOX at their respective predetermined IC_90_ concentrations are used in experiments. TS formation represents a sensitive and specific cellular marker for stem cells. The data presented in Figure 1A illustrate the comparison of tumor spheroid formation in drug-sensitive and drug-resistant phenotypes. The resistant phenotypes exhibit a substantial increase in tumor spheroid number relative to their sensitive counterparts.

The data presented in Figure 1B–D illustrate the expression status of select molecular markers for stem cells. These markers include CD44, OCT-4 and NANOG. Comparison of the expression pattern in drug-sensitive and drug-resistant stem cell phenotypes demonstrates a substantial increase in favor of TAM-R, LAP-R and DOX-R stem cells, respectively.

### 3.2. Colon

The drug-resistant stem cells are maintained in the presence of SUL and 5-FU at their respective predetermined IC_90_ concentrations for experiments. The data presented in Figure 2A illustrate TS formation in the SUL-R phenotype from the 850^MIN^ COL cells, a model for FAP. The TS number is increased in the SUL-R phenotype. The data presented in Figure 2B demonstrate that the expression of the stem cell markers CD44, CD133 and c-Myc is increased in the SUL-R stem cell phenotype.

The data presented in Figure 3A,B illustrate the expression of TS and CD44, CD133 and c-Myc in the Mlh_1_/1638N COL cells, a model for HNPCC. The 5-FU-R stem cells exhibit increased expression of the stem cell markers relative to the 5-FU-S cells [28].

## 4. Experimental Modulation

Transgenic animal models expressing RAS, Myc and HER-2 oncogenes in the mammary epithelium develop mammary cancer. Stable expression of these oncogenes in non-tumorigenic mammary epithelial cells induces tumorigenic transformation [34,35,36,37].

Genetically predisposed mice exhibit germline mutations in the tumor suppressor gene Apc and in DNA mismatch repair genes Mlh1, Msh2 and Msh6. These genetic defects in mice lead to the formation of intestinal adenoma. The 850^MIN^/+ model has been utilized to examine the combinatorial chemo-preventive efficacy of pharmacological agents and natural products [27,28,29].

Naturally occurring dietary agents have documented preventive efficacy in transgenic models expressing RAS, Myc and HER-2 oncogenes in the mammary epithelium and leading to the formation of premalignant lesions such as ductal carcinoma in situ and lobular carcinoma in situ for breast cancer [30,31,32,33].

Naturally occurring phytochemicals such as polyphenols, isoflavones and terpenoids induce anti-proliferative and pro-apoptotic effects via distinct mechanisms in 184-B5/HER cells, a model for HER-2-enriched breast cancer. Mechanistically, growth inhibition is associated with the arrest of cells in the G_1_ phase of the cell cycle and the inhibition of phosphorylated HER-2 expression. Induction of cellular apoptosis is associated with the downregulation of BCL-2 and upregulation of BAX [40].

Chinese herbs, many of them nutritional in nature, have been traditionally used for the management of general health concerns and also for estrogen-related health issues, including breast diseases in women [41,42]. These herbs lack evidence of systemic clinical toxicity. Because of their nontoxic nature and documented human use, these herbs may represent testable therapeutic alternatives. Herbal formulations prescribed in traditional Chinese medicine are commonly prepared as a decoction by boiling a mixture of herbs in water. It is, therefore, conceivable that water-soluble constituents may represent active agents responsible for the efficacy of the herbs. To simulate patient consumption, non-fractionated aqueous extracts of the Chinese herbs have been used to examine their growth-inhibitory efficacy in cellular models of the Luminal A and TNBC subtypes. In the Luminal A model, the growth-inhibitory effects are associated with altered cellular metabolism of estradiol, generating anti-proliferative metabolites [20,21,22]. In the TNBC model, the growth-inhibitory effects are associated with the inhibition of abnormal RB signaling and inhibition of RAS, PI3K and AKT-mediated cell survival pathways [23,24,25,26].

Small molecules targeting programed cell death protein-1, programed death ligand-1, poly ADP-ribose polymerase and antibody–drug conjugates represent novel agents selective for several actionable molecular targets [43].

Recently published evidence suggests that Chinese nutritional herbs and constituent active components [25,41,42], dietary phytochemicals and natural products [44,45,46] may also affect cancer stem cell growth.

### 4.1. Breast: For These Experiments, the Natural Products Vitamin A Derivative ATRA and Terpene CSOL Are Used at Their Predetermined Maximally Cytostatic Concentrations

The data presented in Figure 4A,B illustrate that vitamin A derivative ATRA and the natural terpenoid CSOL inhibit TS formation (Figure 4A) and the expression of CD44, NANOG and OCT-4 (Figure 4B) in the LAP-R stem cell model for HER-2-enriched breast cancer.

### 4.2. Colon

In the 850^MIN^ COL model for familial adenomatous polyposis (FAP) syndrome, independent treatment with mechanistically distinct pharmacological agents and naturally occurring dietary compounds as single agents inhibits the growth of colonic epithelial cells. For example, the pharmacological compounds difluoro-methylornithine, sulindac and celecoxib function as selective inhibitors of ornithine decarboxylase and of cyclooxygenases, respectively. The naturally occurring tea polyphenol epigallocatechin gallate predominantly affects EGFR signaling. Collectively, these agents function as potent inhibitors of the Wnt/APC/β-catenin signaling cascade via the expression of select downstream target genes. Treatment with low-dose combinations of these agents results in a greater reduction in AI colony number, relative to treatment with individual single agents. The superior efficacy of the combinatorial treatment is predominantly due to synergistic interactions via distinct mechanisms [27,28,29]. In addition, several naturally occurring phytochemicals, such as curcumin, epigallocatechin gallate, sulforaphane, resveratrol and genestein, are known to target cancer stem cells [46].

In traditional Chinese medicine, nutritional herbs have also been used for the treatment of colon cancer. Extracts prepared from the roots, leaves, fruits and seeds of the herbs function as potent immunomodulatory and anti-inflammatory agents [47]. In a colon cancer model of patient-derived tumor xenografts, herbal extracts have been shown to enhance the anti-cancer activity of the chemotherapeutic drugs irinotecan and oxaliplatin [48]. At the mechanistic levels, the herbal extracts function as potent inhibitors of receptor tyrosine kinase, mitogen-activated protein kinase-extracellular receptor kinase (MAPK-ERK) and the nuclear factor-kB (NF-kB) signaling pathway [49].

For these experiments, the cells were treated with predetermined maximum cytostatic concentrations of non-fractionated aqueous extracts prepared from the nutritional herbs *Pseudo ginseng* (PC), *Radix salviae* (RS), *Radix paeoniae* (RP) and *Morinda officinalis* (MO). The data presented in Figure 5 demonstrate that treatment with the nutritional herbs inhibited AI colony formation in the 850^MIN^ COL model.

Several natural products, including polyphenols, terpenes and N-3 PUFA, exhibit inhibitory effects in colon cancer models. For these experiments, the cells are treated with natural products at their respective predetermined maximum cytostatic concentrations. The data presented in Figure 6 demonstrate that these natural products are effective in inhibiting TS formation in the SUL-R 850^MIN^ COL model.

For these experiments, the cells are treated with the natural products CUR, the bioactive agent from turmeric, and ATRA, a vitamin A derivative, at their respective predetermined maximum cytostatic concentrations. The data presented in Figure 7A,B illustrate that CUR inhibits TS formation (Figure 7A) and the expression of CD44, CD133 and c-Myc (Figure 7B) in SUL-R 850^MIN^ COL cells, a model for FAP.

## 5. Stem Cell Models—Overview

The multi-step progression of breast and colon cancers is profoundly influenced by a complex interaction of dietary, endocrine and environmental factors. These aspects present formidable challenges for the selection of appropriate therapeutic options. In addition, the emergence of drug-resistant cancer stem cells facilitates the metastatic progression of the disease. Table 5 provides an overview of conventional and targeted therapeutic options and their limitations, developed drug-resistant stem cell models, natural products and their stem-cell-targeting efficacy, as well as future research directions.

## 6. Conclusions

This review discusses the conceptual, phenomenological and mechanistic evidence for cancer stem cell models developed from epithelial cell lines derived from human breast carcinoma and from colonic epithelial cell lines derived from female mice genetically predisposed to intestinal carcinogenesis. These models exhibit relevance to molecular subtypes of clinical breast and colon subtypes. Based on the evidence for the in vivo combinatorial efficacy of pharmacological agents and natural products in the 850^MIN^/+ animal model for FAP syndrome, the evidence of the stem-cell-targeted growth-inhibitory effects of natural products provides a proof of concept that naturally occurring compounds may represent testable therapeutic alternatives for therapy-resistant breast and colon cancer. Collectively, this evidence provides a basis for expanding the present research for its potential clinical relevance.

## 7. Future Directions

The premalignant cancer-initiating stem cells are endowed with resistance to chemotherapy and EMT. These chemo-resistant, EMT-positive cells progress to the metastatic phenotype, which is capable of invading distant metastatic sites [50,51,52,53,54]. In addition, these stem cells exhibit activation of RAS-MAPK-ERK, PI3K, AKT, mTOR and STAT-3/NFkB-mediated survival pathways, which are susceptible to the inhibitory efficacy of mechanistically distinct natural products and nutritional herbs [55,56,57,58,59,60,61,62].

It is notable that the LAP-R, SUL-R and 5-FU-R phenotypes represent relevant models of premalignant cancer-initiating stem cells for HER-2-enriched, FAP and HNPCC cancer subtypes, respectively. Collectively, these stem cell models provide valuable experimental approaches for the genomic, transcriptomic, proteomic and metabolomic analysis of molecular mechanisms that are relevant to the progression of the target premalignant cells to the invasive cancer phenotype.

Breast-carcinoma-derived immortalized cell lines and spontaneously immortalized colonic epithelial cell lines express telomerase enzyme, which represents a specific and sensitive marker for the immortalized phenotype. Synthetic small molecules and naturally occurring phytochemicals that function as telomerase inhibitors may also represent novel testable alternatives [63].

In the estrogen receptor (ER) expressing the Luminal A breast cancer subtype, the ER functions as a ligand-regulated transcription factor. The ER isoforms α and β are responsible for distinct positive and negative growth regulatory functions [8,9,10,11]. Some Chinese nutritional herbs with documented estrogenic properties also function as potent ER-β agonists [22,41,42,47]. These herbs may exhibit growth-inhibitory effects via the upregulation of ER-β in relevant breast cancer models.

The experimental evidence discussed in the present review is derived predominantly from preclinical animal models in vivo, and from established epithelial cell lines in vitro. Collectively, these aspects represent important research directions for future investigations. However, it needs to be recognized that experiments on cancer stem cell models that are derived from established cell lines depend on extrapolation for their clinical relevance. These approaches have provided valuable mechanistic data. However, the potential clinical translatability of these data is strongly dependent on extrapolation.

In an effort to reduce the need for extrapolation, future investigations on patient-derived tumor samples may be of considerable importance. These investigations may include ex vivo models developed from patient-derived tumor xenografts [64], patient-derived organoids [65,66,67,68,69], patient-derived induced pluripotent stem cells [70] and patient-derived ex vivo organ cultures [71]. These approaches provide scientifically robust rationale for the clinical relevance and translatability of preclinical data.

## Figures and Tables

**Figure 1 ijms-23-07055-f001:**
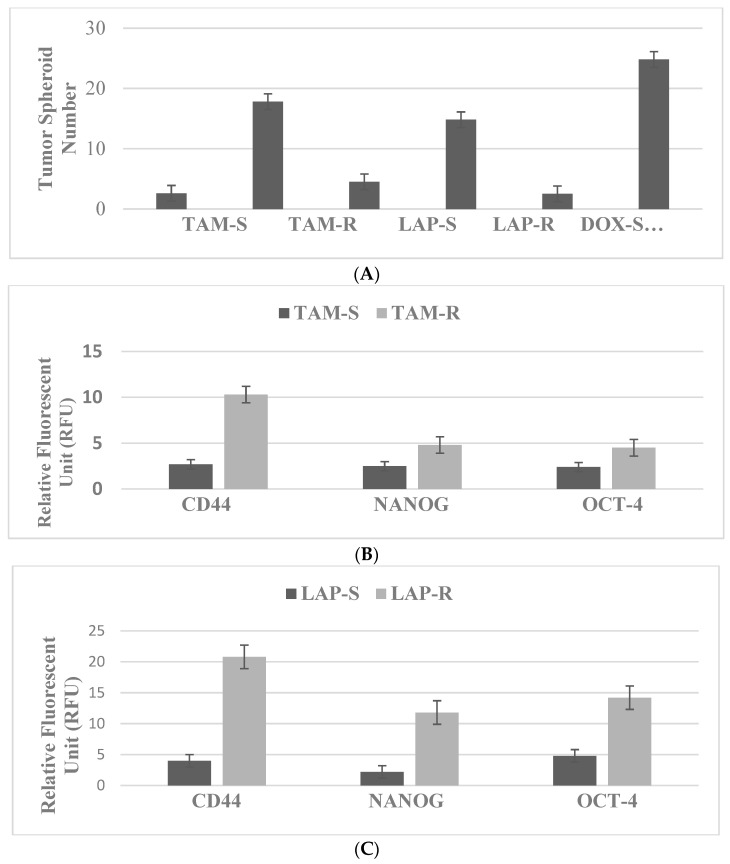
(**A**) Status of tumor spheroid formation. In the presence of cytotoxic drugs, the drug-resistant (R) phenotypes exhibit increased numbers of tumor spheroids relative to their respective drug-sensitive (S) counterparts. TAM, tamoxifen; LAP, lapatinib; DOX, doxorubicin. Status of stem cell markers in drug-resistant phenotypes. (**B**) The expression of CD44, NANOG and OCT-4 in the resistant TAM-R phenotype is increased relative to the sensitive TAM-S phenotype. (**C**) The expression of CD44, NANOG and OCT-4 is increased in the resistant LAP-R phenotype relative to the sensitive LAP-S phenotype. (**D**) The expression of CD44, NANOG and OCT-4 is increased in the resistant DOX-R phenotype relative to the sensitive DOX-S phenotype. CD44, cluster of differentiation 44; NANOG, homeobox transcription factor; OCT-4, octamer-binding transcription factor-4; TAM, tamoxifen; LAP, lapatinib; DOX, doxorubicin.

**Figure 2 ijms-23-07055-f002:**
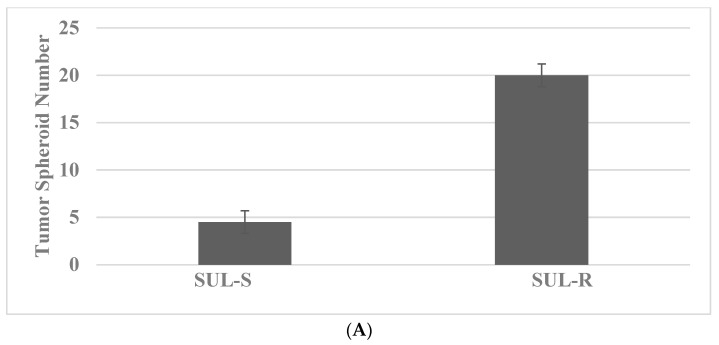
Status of tumor spheroid formation and stem cell marker expression in 850^MIN^ COL model. (**A**) The tumor spheroid number is increased in the resistant SUL-R phenotype relative to the sensitive SUL-S phenotype. (**B**) The expression of CD44, CD133 and c-Myc is increased in the resistant SUL-R phenotype relative to the sensitive SUL-S phenotype. SUL, sulindac; CD44, cluster of differentiation 44; CD133, cluster of differentiation 133; c-Myc, cellular Myc.

**Figure 3 ijms-23-07055-f003:**
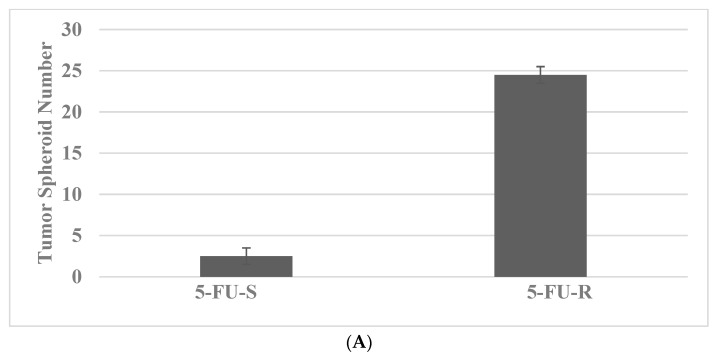
Status of tumor spheroid formation and stem cell marker expression in Mlh_1_/1638N COL model. (**A**) The tumor spheroid number is increased in the resistant 5-FU-R phenotype relative to the sensitive 5-FU-S phenotype. (**B**) The expression of CD44, CD133 and c-Myc is increased in the resistant 5-FU-R phenotype relative to the sensitive 5-FU-S phenotype. 5-FU, 5-fluro-uracil; CD44, cluster of differentiation 44; CD133, cluster of differentiation 133; c-Myc, cellular Myc.

**Figure 4 ijms-23-07055-f004:**
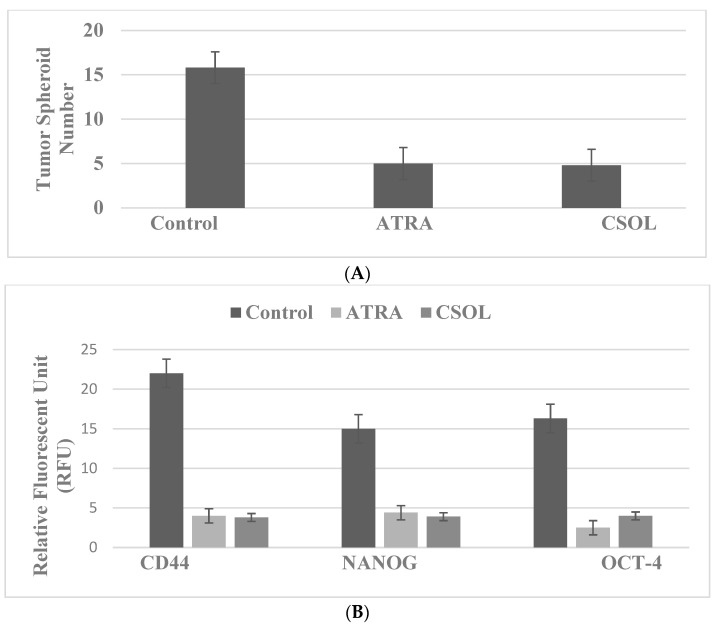
Inhibition of stem cell marker expression in the LAP-R 184-B5/HER model. (**A**) Treatment with ATRA and CSOL decreased tumor spheroid number relative to the solvent control. (**B**) Treatment with ATRA and CSOL decreased the expression of CD44, NANOG and OCT-4 relative to the solvent control. ATRA, all-trans retinoic acid; CSOL, carnosol; CD44, cluster of differentiation 44; NANOG, homeobox transcription factor; OCT-4, octamer-binding transcription factor-4.

**Figure 5 ijms-23-07055-f005:**
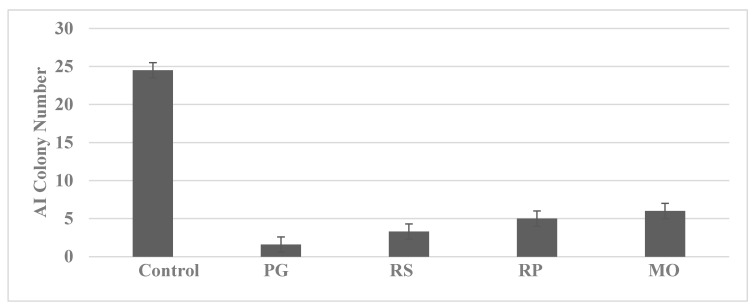
Inhibition of AI colony formation by nutritional herbs. PG, *Pseudo ginseng*; RS, *Radix salviae*; RP, *Radix paeoniae*; MO, *Morinda officinalis*.

**Figure 6 ijms-23-07055-f006:**
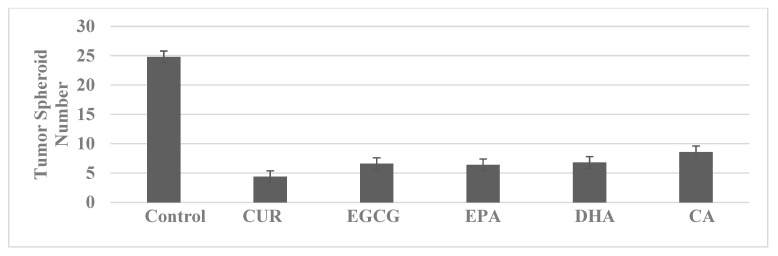
Inhibition of tumor spheroid formation by phytochemicals. CUR, curcumin; EGCG, epigallocatechin gallate; EPA, eicosapentaenoic acid; DHA, docosahexaenoic acid; CA, carnosic acid.

**Figure 7 ijms-23-07055-f007:**
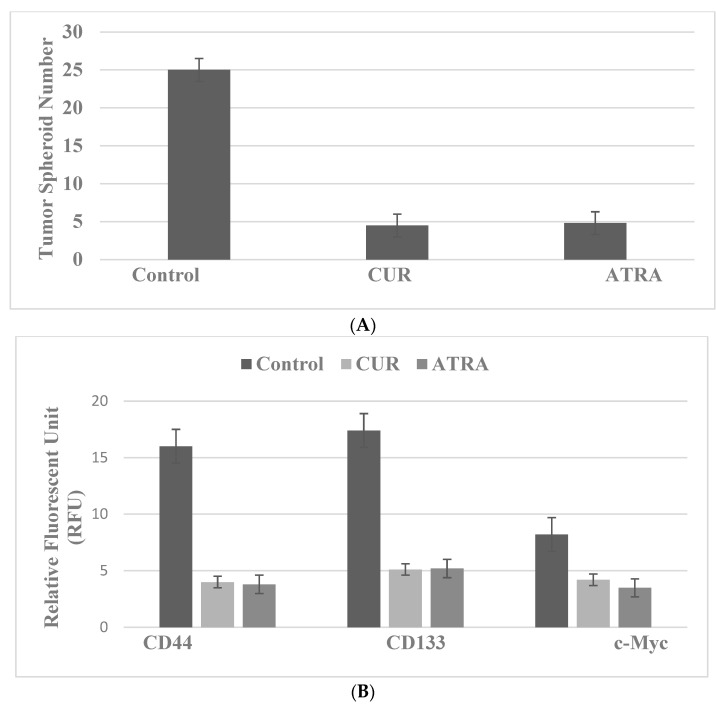
Inhibition of stem cell markers in the SUL-R 850^MIN^ COL model. (**A**) Treatment with CUR and ATRA decreased tumor spheroid number relative to the solvent control. (**B**) Treatment with CUR and ATRA inhibited the expression of CD44, CD133 and c-Myc relative to the solvent control. CUR, curcumin; ATRA, all-trans retinoic acid; CD44, cluster of differentiation 44; CD133, cluster of differentiation 133; c-Myc, cellular Myc.

**Table 1 ijms-23-07055-t001:** Cellular models for breast and colon cancer.

Model	Molecular Characteristics			Origin	Clinical Subtype
Breast	ER	PR	HER-2		
184-B5	-	-	-	Reduction mammoplasty	Female normal breast
MCF-7	+	+	-	Female breast carcinoma	Luminal A
184-B5/HER	-	-	+	HER-2 positive	HER-2-enriched
MDA-MB-231	-	-	-	Female breast carcinoma	TNBC
**Colon**	**Apc**		**Mlh1**		
C57 COL	+/+		+/+	Female mouse colonic epithelium	Normal colon
850^MIN^ COL	+/−		+/+	Female mouse Apc codon 850 mutation	FAP
Mlh_1_/1638N COL	+/−		−/−	Female mouse Apc codon 1638N mutation, Mlh1 allelic deletion	HNPCC
HCT-116	**Apc**		**β-Cat.**	Somatic mutation	Male sporadic colon cancer
	WT		MT		
SW480	MT		WT	Somatic mutation	Male sporadic colon cancer

ER, estrogen receptor; PR, progesterone receptor; HER-2, human epidermal growth factor receptor-2; Apc, adenomatous polyposis coli; Mlh1, Mut-L; β-cat., β-catenin; FAP, familial adenomatous polyposis; HNPCC, hereditary non-polyposis colon cancer; WT, wild type; MT, mutant.

**Table 2 ijms-23-07055-t002:** Growth characteristics of cellular models for breast cancer.

Endpoint	Cellular Model			
	184-B5	MCF-7	184-B5/HER	MDA-MB-231
AI Colony Formation	0/18	18/18	18/18	18/18
Incidence	0%	100%	100%	100%
Colony Number	ND	30.9 ± 2.4	23.0 ± 2.6	38.9 ± 1.6
Tumor Formation	0/10	10/10	10/10	10/10
Incidence	0%	100%	100%	100%
Latent Period (weeks)	24	3–5	3–5	3–5

HER, human epidermal growth factor receptor; AI, anchorage-independent; ND, not detectable.

**Table 3 ijms-23-07055-t003:** Growth characteristics of cellular models for colon cancer.

Endpoint	Cellular Model		
	C57COL	850^MIN^COL	Mlh_1_/1638COL
AI Colony Formation	0/18	18/18	16/18
Incidence	0%	100%	88.9%
Colony Number	ND	18.9 ± 2.5	15.2 ± 1.4
Tumor Formation	0/10	8/10	6/10
Incidence	0%	80%	60%
Latent Period (weeks)	24	3–5	3–5

MIN, multiple intestinal neoplasia; AI, anchorage-independent.

**Table 4 ijms-23-07055-t004:** Pharmacological agents for isolation of resistant phenotype.

Agent	Type	Molecular Target	Stem Cell Model
**Breast**			
TAM	SERM	ER	MCF-7 TAM-R, Luminal A
LAP	Small-molecule inhibitor	EGFR, HER/2	184-B5/HER LAP-R, HER-2-enriched
DOX	DNA inhibitor	S-phase inhibition	MDA-MD-231, DOX-R, TNBC
**Colon**			
SUL	NSAID	COX-1. COX-2	850^MIN^COL SUL-R, FAP
5-FU	DNA inhibitor	S-phase inhibition	Mlh_1_/1638N COL, 5-FU-R, HNPCC

TAM, tamoxifen; SERM, selective estrogen receptor modulator; LAP, lapatinib; EGFR, epidermal growth factor receptor; HER-2, human epidermal growth factor receptor-2; DOX, doxorubicin; SUL, sulindac; NSAID, non-steroidal anti-inflammatory drug; COX-1/COX-2, cyclooxygenase-1, cyclooxygenase-2; TNBC, triple-negative breast cancer; FAP, familial adenomatous polyposis; 5-FU, 5-fluoro-uracil; HNPCC, hereditary non-polyposis colon cancer.

**Table 5 ijms-23-07055-t005:** Stem cell models: therapeutic alternatives.

Therapeutic Options	Stem Cell Models	Testable Alternatives
Breast		
Multi-drug chemotherapy	Drug resistance	Dietary phytochemicals, nutritional herbs, micro-nutrients
Endocrine therapy	TAM-R, LAP-R, DOX-R	LAP-R: ATRA, CSOL
Molecular-targeted therapy	Markers: TS, CD44, NANOG, OCT-4	Marker downregulation
**Selective inhibitors:** SERM, SERD, AI, PI3K, AKT, m TOR		
**Colon**		
Multi-drug chemotherapy	Drug resistance	
Molecular-targeted therapy	SUL-R, 5-FU-R	SUL-R: CUR, EGCG, EPA, DHA, CA, ATRA
**Selective inhibitors:** EGFR, ODC, COX-2, NSAID	Markers: TS, CD44, CD133, c-Myc	Marker downregulation
**Limitations:** Systemic toxicity, therapy resistance, drug-resistant stem cell population		**Advantages:** Documented human consumption, low systemic toxicity
**Future research:** PDTX, PDTO, stem cell models for clinical cancer		**Future research:** Bioactive constituents from nutritional herbs

SERM, selective estrogen receptor modulator; SERD, selective estrogen receptor degrader; AI, aromatase inhibitor; PI3K, phosphotidyl-inositol 3 kinase; AKT, protein kinase B; m TOR, molecular target of rapamycin; EGFR, epidermal growth factor receptor; ODC, ornithine decarboxylase; COX-2, cyclooxygenase-2; NSAID, non-steroidal anti-inflammatory drug; PDTX, patient-derived tumor xenografts; PDTO, patient-derived tumor organoids; TAM-R, tamoxifen-resistant; LAP-R, lapatinib-resistant; DOX-r, doxorubicin-resistant; TS, tumor spheroid; CD44, cluster of differentiation 44; NANOG, homeobox transcription factor; OCT-4, octamer-binding transcription factor-4; SUL-R, sulindac-resistant; 5-FU-R, 5-fluoro-uracil-resistant; CD133, cluster of differentiation 133; c-Myc, cellular Myc; ATRA, all-trans retinoic acid; CSOL, carnosol; CUR, curcumin; EGCG, epigallocatechin gallate; EPA, eicosapentaenoic acid; DHA, docosahexaenoic acid; CA, carnosic acid.

## Data Availability

The data sets used in the present review are available from the author on reasonable request.

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
