# Peer review of "Stem Cell Models for Cancer Therapy"

_ijms, 2022, doi:10.3390/ijms23137055_

Round 1
Reviewer 1 Report
This review by Dr. Telang on Stem cell models for cancer therapy is well-written and provides a clear overview of the therapeutic alternative offered by naturally-occurring dietary phytochemicals.
This reviewer has no specific concerns, and enjoyed reading the manuscript except for the 20 self-citations (out of 73 references). If possible, it is best to avoid such high number of self-citations.
Author Response
I thank the reviewers for their insightful critique and helpful recommendations on the originally submitted manuscript. I have carefully considered all the recommendations and have revised the manuscript. Newly added segments in the text of the revised version of the manuscript are highlighted in bold face.
The following is my point by point response to concerns/issues raised by the reviewers.
Reviewer 1:
Concern: High number of self-citations.
Response: In the present review originally published primary data has been summarized to discuss relevant evidence in a concise manner. Therefore, it was important to cite original reference for the benefit of the reader. Note that 20 self-cited references out of a total of 73, represents only about 27% of the total references. However, pursuant to the recommendation, I have attempted to reduce the number of self-cited references. The revised reference section now includes only those references that are essential to the subject matter being discussed. The reference numbers in the text have been appropriately corrected.
Reviewer 2 Report
Telang Nitin T provides a review of the stem cell models for cancer therapy. The manuscript aims to provide a systematic discussion on the conceptual and experimental aspects relevant to chemo-endocrine therapy of breast and colon cancer, molecular aspects of cancer stem cells, and potential stem cell-targeting lead compounds as testable alternatives against the progression of therapy-resistant breast and colon cancer.
Despite the review address a very interesting topic in oncology research, but the submitted manuscript lacks a logical and comprehensive description of what has been recently published in the literature, jumping from paragraph to paragraph without a logical connection between the different points that are mention. Therefore, a more logical and better-structured form will definitely be a better form to present an overview of the topic. For example, the Experimental models topic is first mentioned the chemical carcinogens, environmental and dietary factors, and germline mutations in BRCA1, BRCA2 and APC (which must be written in italic) without presenting a logical between the different points
Is not clear to the reader the origin of the data presented in Figures 1 to 7. Is this original data? Taken from previous publications? If the presented data refers to original data which conditions were used in drug concentrations? Cell lines? etc?
If the data presented refers to original data what is the rationale to present 7 figures of original data on a review?
Several sentences in the manuscript need to be re-written due to the lack of sense or typo mistakes
Examples
In the abstract, the sentence “the premalignant cancer stem cells exhibit activation of select cancer cell signaling pathways and undergo epithelial-mesenchymal transition… ” should be revised. The term premalignant is misleading in the sentence.
The following sentence is very difficult to read and understand and should be revised “Natural products such as dietary phytochemicals, micro-nutrients and nutritional herbs have documented cancer preventive efficacy on epithelial organ site cancers of breast and colon in part, via targeting cancer stem cells”
Is not clear want the author means by “epithelial organ sites”
Tables and not formatted in a readable form and it should be corrected.
Author Response
Concern: Review lacks a logical and comprehensive description of published evidence.
Response: The lack of logical and comprehensive description has now been remedied by inclusion of citation of relevant references in the text. Additionally, discussion of experimental evidence is supported by citations of original references for published primary data.
The following is my response to the examples cited by the reviewer. .
Concern: Experimental Models: Multi-factorial cancer progression ----germline mutations’.
Response: This segment is revised to provide a link between exogenous factors and germ line mutations and their impact on genetic predisposition.
Concern: Figures 1-7: Origin of the data is not clear.
Response: The data contained in figures 1-7 are summarized from published primary data. The rationale to include summarized data is to provide relevant mechanistic evidence. The details regarding experimental conditions and drug concentrations used on the drug-resistant stem cell models are now clearly stated in the text.
Concern: Several segments in the text need to be revised to enhance clarity.
Response: Pursuant to the recommendation the following segments in the text are revised.
Abstract: ‘Premalignant cancer stem cells----epithelial-mesenchymal transition’.
Note that the term ‘premalignant’ is used to describe the potential of the stem cells for developing malignant metastasis. This segment is revised to enhance the clarity.
Concern: ‘Natural products----cancer stem cells’.
Response: This segment is revised to eliminate confusion and enhance clarity.
Concern: ‘epithelial organ site’
Response: This term is now deleted
Concern: Tables are not formatted.
Response: The Tables in the text were formatted and the data were properly aligned in the originally submitted manuscript. It is unclear how the formatting and alignment of the content in the Tables has been compromised in the formatted version of the manuscript. All the Tables in the revised version are now properly formatted and the data are aligned. .